# Antioxidants from *Galium verum* as Ingredients for the Design of New Dermatocosmetic Products

**DOI:** 10.3390/plants11192454

**Published:** 2022-09-20

**Authors:** Delia Turcov, Ana Simona Barna, Adriana Trifan, Alexandra Cristina Blaga, Alexandra Maria Tanasă, Daniela Suteu

**Affiliations:** 1Faculty of Chemical Engineering an Environmental Protection “Cristofor Simionescu”, “Gheorghe Asachi” Technical University, 71 A Mangeron Blvd., 700050 Iasi, Romania; 2Faculty of Pharmacy, “Grigore T. Popa” University of Medicine and Pharmacy, Universitatii Street, No. 16, 700115 Iasi, Romania

**Keywords:** *Galium verum*, antioxidant, extraction, dermato-cosmetic product, vegetal extract

## Abstract

The aim of this study was to use plant extracts from spontaneous flora of Moldova (Rediu-Iasi area, Romania) as polyphenols and flavonoids source in order to obtain new dermato-cosmetic formulas to prevent the actions of oxidative stress on skin. Plant extracts (from raw and dried *Galium verum* sp.) were obtained by: cold classical maceration (M), heat reflux extraction (HTE) and ultrasound assisted extraction (UAE). The extracts were characterized by spectrophotometric method (for polyphenols and flavonoids content and for DPPH antioxidant activity evaluation). In order to evaluate the combating and/or attenuating effects of oxidative stress on skin, the study was continued with the elaboration of emulsions that incorporate one of these extracts. The emulsions were preliminarily characterized by determining the stability over time. The obtained results encourage research in the direction of deeper characterization of these emulsions to determine the microbiological stability and dermatological tests performed on the skin treated with these new products.

## 1. Introduction

Although hundreds of years old, traditional medicine shares with modern medicine one of its important components: the source of bioactive ingredients. Due to low side-effects, compared to synthetic medicines, and despite huge amount of research, plants, the oldest form of healthcare known to mankind [1], have not exhausted their supply of bioactive ingredients, useful in maintaining health and improving several health conditions.

With a long history as a traditional healing plant, *Galium* species drew attention for intense investigations in recent years, as they prove to represent a safe, accessible, and efficient natural health remedy. *Galium* sp. belongs to the Rubiaceae family and are perennial herbaceous flowering plants. There are up to 650 species of *Galium* [2] spread all over the world (Africa, Asia, North America, Europe). Over one third of the species are distributed along Europe. The name belongs from the Greek word “gala” which reflects the basic use of this plant in curdling milk in order to obtain cheese. *Galium* sp. represent some of the plants widely used in traditional medicine in an impressive range of internal and external conditions. However, scientific studies dedicated to full knowledge of this plant and its optimum uses are not so numerous compared to other plant families. In recent years, scientific research of *Galium* sp. have been intensified, especially in areas with indigenous spontaneous flora rich in *Galium* sp: Serbia, Slovakia, Romania, and Ukraine [3,4,5].

The species reported in scientific literature are: *G. verum*, *G. mollugo* (hedge bedstraw, false baby’s breath), *G. album*, *G. rivale*, *G. pseudoaristatum*, *G. purpureum*, *G. aparine* (cleavers, bedstraw, catchweed, goosegrass).

In Romania about 38 species have been identified, six of them with yellow flowers [2,4,6,7,8]. Among these there are investigations and mentions of: *G. verum* (lady’s bedstraw); *G. mollugo* L. (hedge bedstraw); *G. aparine* L. (cleavers or stickyweed); *G. odoratum* L. (syn. Asperula odorata, Scop. Or woodruff); *G. rivale*; *G. pseudoaristatum*; *G. purpureum* (syn. Asperula purpurea, reddish bedstraw); *G. ruboides* L.; *G. schultestt*
*West* (forest bedstraw); *G. album Mill*.

Two species are well known, *Galium verum*, with yellow inflorescence and *Gallium mollugo*, with white inflorescence. In therapy, there are three species most frequently used: *Galium aparine* L., *Galium odoratum* L., and *Galium verum* L. *Galium verum*’s first traditional medical uses were as a choleretic, diuretic or spasmolytic treatment. There are detailed research showing a complex and rich content of phytochemicals in flowers and leaves of *Galium verum*, with the highest total phenolic and flavonoid composition, alongside *G. purpureum*. The major constituents identified belong to iridoid glycosides terpenes, phenolic acids, flavonoids, polysaccharide complex, aldehydes, alcohols, anthraquinones, acids, small amounts of tannins, saponines, waxes, pigments, vitamin C and essential oils [1,5,6,9,10].

Due to beneficial effects of *Galium* species noticed over time, there are now galenic remedies and dietary supplements considered to help in several health disorders, mainly involving immune system, anti-inflammatory processes, detoxication and oxidative stress. Considering the species widely spread in areas of our interest, across Romania, the reported biological activities of *Gallium verum* are presented in Table 1.

## 2. Methods for Obtaining and Characterizing Extracts from *Galium* Species

The extraction protocols are continuously optimized in order to obtain a high quality, safety, and efficient extraction through a fast, cost effective, comprehensive, and reproducible procedure. There are some extraction techniques investigated so far for *Galium* species [7,9,11,12]: (i) Maceration in hydro-alcoholic solution with 20%, 30%, 50%, 60%, 70% or 96% ethanol for 7 days or methanol; (ii) Percolation in methanol; (iii) Ultra-sound assisted extraction; (iv) Reflux extraction; (v) Dispersive liquid-liquid phase microextraction with NaCl and natural deep eutectic solvent medium and ultrasound assisted. The study and application of these extraction techniques is based on a series of variations of the working parameters, such as the solid ratio: liquid, type of solvent, volume and concentration of the extraction agent, extraction time.

Various physic-chemical methods have been used for the qualitative and quantitative characterization of *Galium* extracts, such as: chromatographic analysis (HPTLC, HPLC) [9]; HPLC assisted by MS detection for chlorogenic and caffeic acids [6]; LC/MS-MS quantitative analyze of some flavones; Spectroscopy assay-Electron Paramagnetic Resonance (EPR) [9]; HPLC-DAD-MS/MS [12]; UHPLC [12]; HPLC–PDA [7].

Along with these methods, a number of specific methods have been used, in accordance with the direction of subsequent use of biologically active extracts and/or compounds, such as: antioxidant evaluation (cell-free models: against H_2_O_2_, NO species, (DPPH), Trolox equivalent antioxidant capacity (TEAC), total polyphenols content (TPC)) [9,12], in vivo-experiments in female rats, biochemical analyses [9]; Statistical measurements [9]; Reaction of the lymphocyte blast transformation (RLBT) for testing the immunomodulatory activity [12]; and Immunomodulatory potential of *G. aparine* (highest in aqueous extract compounds) [12].

The aim of this study was to obtain new ecological and effective products in the fight against antioxidant stress on the skin, which would contribute to improving and ensuring the quality of people’s lives. In this sense, a series of dermatocosmetic emulsions with *Galium verum* extract as active ingredient were prepared and characterized. The extract was obtained using different liquid-solid extraction methods (maceration (M), thermal reflux extraction (HTE) and ultrasound-assisted extraction (sono-extraction-UAE). The efficiency of these methods was evaluated according to a series of parameters (solid/liquid ratio, extraction time and extraction solvent concentration) by calculating the extraction yield. The extracts obtained were characterized by the quantitative determination of the main classes of biologically active compounds extracted. The emulsions obtained were studied for their stability following a series of parameters over time (pH, conductivity, phase homogeneity, microbiological stability).

## 3. Results and Discussions

### 3.1. Evaluation of the Efficiency of the Extraction Techniques Used

As extraction methods, maceration, heat reflux extraction, and ultrasound assisted extraction were used. Their efficiency was appreciated using the extraction degree, which was calculated according to the parameters established for each technique, such as: solid/liquid ratio, contact time, plant condition (green or dry) and solvent extraction composition. The results obtained for each form of plant processing: green and dried, are shown in Table 2 and Table 3. For this purpose, the plant extracts were processed to obtain residues by complete evaporation of 5 mL of the sample extract in a thermostatic oven Poleko SLW53 model (Pol-Eko-Aparatura sp.j., Wodzisław Śląski, Poland) (temperature up to 50 °C).

The analysis of the experimental data obtained and presented in Table 2 and Table 3 clearly emphasizes that the extraction process studied depends on the condition of the plant used for extraction (dry or fresh), and on the extraction technique applied. Within the same technique, the solid/liquid ratio, the extraction time and the extractant concentration are also important. It is very easy to notice that the solvent with the best results, regardless of the extraction technique but also considering the criterion of economic efficiency, proves to be 50% ethanol concentration.

The best extraction yields were obtained in the case of the dried plant, which is observed from almost double yields compared to the case of the wet plant, in the same conditions. A first explanation would be that the dried plant (with 8.4% humidity) is used in larger quantities in samples, because the fresh plant has a considerable amount of water (69.25%) which is lost by drying, together with a quantity from more volatile chemical compounds.

For both types of plants, in general, the order of efficiency of the extraction techniques used is as follows: reflux extraction, maceration, and ultrasonic assisted extraction.

In the case of the same extraction technique, it is obvious the dependence of the extraction yield on the concentration of the solvent used for extraction, on the extraction time, as well as on the solid/liquid ratio used. Thus, in the case of Maceration (M), maximum yields are registered between 34.4% (S/L + 1:15; 4 days; solvent 50%) and 33.75% (S/L + 1:15; 11 days; solvent 50%) in the case of the dried plant and between 14.28% (S/L = 1:30; 8 days; solvent 70%) and 12.57% (S/L = 1:30; 8 days; solvent 50%) in the case of the green plant. When Heat Reflux Extraction (HTE) was used, maximum yields were obtained between 34.38 (S/L = 1:20; 60 min; solvent 70%) and 33.59% (S/L = 1: 0; 60 min; solvent 30%), respectively 33.6% (S/L = 1:20; 30 min; solvent 50%) when using the dried plant and between 15.59% (S/L = 1:20; 30 min; solvent 50%) and 14.4% (S/L = 1:15; 60 min; solvent 50%), respectively 14.38% (S/L = 1:20; 60 min; solvent 70%) in the case of the green plant. Using the Ultrasound Assisted Extraction (UAE) the following maximum yields were obtained: 17.6% (S/L = 1:20; 6 min; solvent 50%) and 16.19% (S/L = 1:30; 4 min; solvent 50%) for the dried plant and between 6.02% (S/L = 1:30; 4 min; solvent 50%) and 3.19% (S/L = 1:20; 4 or 2 min; solvent 30% or 50%) in the case of the green plant.

From the point of view of the extraction yield obtained, we can conclude that in the case of the dried plant the best yields are obtained. In addition, the use, as extraction methods, of the Extraction with thermal reflux (HTE) in conditions of: solid/liquid ratio of 1:20 and concentration of 50% extraction solvent (ethyl alcohol), respectively of Maceration (M) in conditions of S/L = 1:15, and concentration of 50% extraction solvent (ethyl alcohol), allows to obtain the best results.

Our study continued using the dried *Galium verum* plant.

### 3.2. Total Polyphenol and Total Flavonoids Assay

The results of the experiments performed in duplicate, using for extraction the dried plant of *Galium verum*, were dependent on the extraction technique and their influencing factors, and are presented in Figure 1 and Figure 2. The analysis of the data from Figure 1 regarding the total content of polyphenols allows a series of information depending on the influencing factors considered in the development of the studied extraction method. In the case of different concentrations of the extracting agent, the highest amounts of polyphenols were extracted in the case of the three variants of HTE, the difference being made by the working conditions: 71.539 μg GAE/g in the conditions S/L = 1:20; 60 min and 70% ethyl alcohol concentration, 63.703 μg GAE/g followed by S/L conditions = 1:20; 60 min and 50% ethyl alcohol concentration and 58.307 μg GAE/g under S/L conditions = 1:20; 60 min and 30% ethyl alcohol concentration. Considering the case of the plant: the highest amounts of polyphenols were extracted in the case of HTE: 78.894 μg GAE/g (under S/L conditions = 1:30; 60 min and 50% ethyl alcohol concentration), HTE: 63.703 μg GAE/g (under S/L conditions = 1:20; 60 min and 50% ethyl alcohol concentration); M: 59.669 μg GAE/g (under S/L conditions = 1:30; 8 days and 50% ethyl alcohol concentration). Regarding the extraction time, the highest amounts of polyphenols were also obtained by HTE: 69.089 μg GAE/g in S/L conditions of 1:20, extractant with a concentration of 50% and an extraction time of 90 min; 64.585 μg GAE/g in 1:20 S/L conditions, 50% concentration extractant and an extraction time of 30 min, respectively 63.703 μg GAE/g in 1:20 S/L conditions, 50% concentration extractant, and an extraction time of 60 min.

In conclusion, the best results for polyphenols extraction were obtained by HTE in the following conditions: S/L ratio of 1:30, solvent concentration 50%, and extraction time of 60 min. It was followed by HTE (S/L ratio of 1:20, solvent concentration 70%, and extraction time of 60 min).

In the case of the flavonoid content, the performances of the studied extraction techniques were also analyzed according to the variable parameter considered.

Taking as a criterion the concentration of the extraction solvent, the best results were obtained in the case of HTE (70%)—35.998 mg QE/g followed by maceration (M) with 33.436 mg QE/g. Considering the value of the solid/liquid ratio as an evaluation criterion, the best results were recorded in the case of HTE (1:30/50%/60 min)—37.984 mg QE/g, followed by maceration (1:15/50%, 8 days). When the extraction time was the criterion of appreciation, the best results were obtained in the case of M (8 days) with 33.436 mg QE/g followed by HTE (1:20/50%/90 min) with 32.481 mg QE/g.

In conclusion, in the case of flavonoid content, the best results were obtained when using HTE extraction (37.984 mg QE/g) under S/L ratio conditions of 1:30, solvent concentration of 50% and 60 min extraction time, then HTE (35.998 mg QE/g) in S/L ratio conditions of 1:20, solvent concentration of 70% and extraction time 60 min, equal to maceration (33.436 mg QE/g) in ratio conditions S/L of 1:15, solvent concentration of 50% and extraction time of 8 days.

Regarding the analyzed plant extracts: (1) the amount of flavonoids is higher than that of polyphenols. This observation is also in agreement with other studies in the literature that specify a greater qualitative and quantitative variety of flavonoids (for example Rutin, Quercetin and its derivatives, Kaempferol) compared to polyphenols (the easiest and most frequently identified being Rosmarinic acid) [3]; (2) the extraction method that led to the highest amounts of extracted bioactive compounds is reflux extraction (HTE), followed by maceration (M); (3) applying the HTE technique in S/L ratio conditions of 1:30 and 50% extraction solvent concentration, the highest amounts of both flavonoids (37.984 mg QE/g) and polyphenols (78.894 μg GAE/g) are obtained.

### 3.3. Determination of Antioxidant Activity

To evaluate the antioxidant capacity, the extract from *Galium verum* dried plant (R3 sz) was selected, obtained by the method that presented the best extraction yield (HTE-reflux at temperature: S/L = 1:20; EtOH 70%). For comparison, we tested the green plant extract (R3 cr), obtained under the same conditions. The results obtained are presented in Table 4.

Table 4 also contains the results of the determinations regarding the content of flavonoids and polyphenols, analyzes performed for the selected extracts. It is observed that the values obtained are in line with the trend of the results presented in Figure 1 and Figure 2, respectively, for large quantities in the case of using the dried plant. These higher amounts also lead to higher antioxidant activities.

### 3.4. Preliminary Evaluation of Dermatocosmetic Emulsions Containing Galium verum Extracts

A criterion for assessing the quality of a dermato-cosmetic preparation is that of stability in different conditions. In this sense the stability of the obtained dermato-cosmetic emulsions was evaluated preliminary through a series of physical analyses, performed at three moments from the preparation: at 24 h after the formulation, after 7 days and 21 days of storage at ambient temperature in the emulsion container. The following physical analyzes were performed: pH determination, centrifuge and vortex tests, sensory evaluation (odor, color, texture, and general aspect), conductivity measurement, and study of microscopic images [13,14]. The results obtained are summarized in Table 5.

#### 3.4.1. pH Determination

It is known that the pH of the skin varies between 5.6 and 5.5. The pH measurements performed on the prepared emulsions indicate that they do not show variations during storage (4.99 to 5.04) (Table 5). This aspect can be attributed to the stable conductivity values but also to the absence of any chemical degradation, so that the emulsions can be considered suitable for topical application and acceptable to avoid the risk of irritation when applied to the skin.

#### 3.4.2. Phases Separation

The stability of the component phases of the studied system was studied under the action of centrifugal and vibrational force. When the operations were completed, the formulations were examined for identification of the phase’s behavior. The results observed are summarized in Table 5.

#### 3.4.3. Microscopic Images

Microscopic images were taken from samples of the emulsions prepared after 7 days of storage under normal temperature conditions. The images (Table 5) show the uniformity of the sample, suggesting the maintenance of compatibility and phase homogeneity during the storage period.

#### 3.4.4. Conductivity Measurements

Conductometric analysis was performed to identify the behavior of the active substance (plant extract) trapped in the inner phase of the primary emulsion. It is known that the more an electrolyte is released, the active substance is free to move in the external aqueous phase, and the less sustained the effects will be. If during storage the conductivity values increase this would be due to the diffusion of the active substance, the coalescence of the internal and aqueous phases, or the destruction of the oil film due to the osmotic pressure and the leakage of the internal aqueous phase [13,14,15]. The results of the measurements, systematized in Table 5, show the stability over time of the studied emulsions.

The tests performed regarding the stability of the analyzed emulsions showed that all the formulations presented a good mechanical and physical stability, following study into the chemical and microbiological stability. The incorporation of the natural product did not negatively affect the stability of the studied cosmetic formulations.

## 4. Materials and Methods

### 4.1. Plant Materials

The vegetal material used in this study was represented by *Galium verum* sp. species, harvested in June 2021, Iasi-Rediu geographical area (Romania), during the maximum flowering period. From the plants only the flowers were used for the selected extraction techniques. Some of them were processed immediately after collection by dividing them into small pieces, and another part was dried in a dark and well-ventilated environment and then ground using a food mill (particles size between 2–5 mm). The crushed dried plant was stored in paper bags or brown glass containers for later use. After crushing, the humidity of the samples prepared for analysis was determined using a Kern thermobalance. The humidity of the green samples was 69.25% and of the dry samples was 8.4%.

### 4.2. Extraction Procedure

Extraction was made using conventional and unconventional extraction procedures in the following variants: Heat Reflux Extraction (HTE), room temperature maceration (M), Ultrasound Assisted Extraction (UAE), using a work protocol (extraction time, salt/solvent ratio, and extraction solvent concentrations) established based on our previous studies [16,17] and after some preliminary tests were carried out (data not shown). In all the extraction techniques, 96% ethanol was used to prepare extraction solvent: hydro alcoholic solution with 30%, 50%, 70% *v*/*v* concentration.

The technological flow of obtaining plant extracts consists of two stages: the first stage is represented by the primary processing, which consists of harvesting the plants, drying, grinding, and packing.

The second stage consisted of the extraction of solid-liquid itself and this took place taking into account a number of physical parameters that influence the effectiveness of the process solvent concentration, extraction times and solid/liquid ratio.

In order to evaluate the extraction efficiency, we used the extraction yield. To calculate it, a 5 mL sample of each plant extract was evaporated to dryness (complete drying) at a constant temperature up to 50 °C, using a thermostatic oven. Extraction yield was calculated by the relation:(1)$$\eta \%=\frac{m_{residue}\cdot V_{extract}}{n_{extract}\cdot m_{solid\;sample}}\cdot 100$$
where *m_residue_* represents the mass of the residue obtained after evaporation to dryness, (g); *V_extract_*—the volume of the vegetal extract sample used for evaporation to dryness, (mL); *n_extract_*—the total volume of extract obtained by liquid-solid extraction technique, (mL); *m_solid sample_*—the mass of vegetal powder used in liquid-solid extraction process (g).

### 4.3. Total Polyphenol Content

All experiments for the determination of polyphenols’ content in vegetal extracts were done in duplicate. To evaluate the amount of polyphenolic compounds it was used the Folin-Ciocalteu method [18], using as standard references gallic acid, and the results are given in (μg GAE/mL extract).

The samples were prepared as follows for determinations: Folin-Ciocalteu reagent (0.5 mL) was added to 0.5 mL of *Galium verum* sp. extracts and the system was kept at room temperature for a 5 min and after that 8 mL of a 7.5% aqueous sodium carbonate solution was added and the mixture was incubated for 2 h at room temperature in the dark. The calibration curve method was used to determine the concentration. The standards and samples were recorded using a Shimadzu UV-1280 UV-VIS Spectrophotometer (Shimadzu Corporation, Kyoto, Japan) at the maximum wavelength of 760 nm.

### 4.4. Total Flavonoid Content

All experiments for the determination of flavonoids’ content in vegetal extracts were done in duplicate. The quantification of total flavonoids was realized by spectrophotometric method based on the complexation reaction between flavonoids with aluminum chloride [19,20]. Briefly, 2% aqueous solution of aluminum chloride (1 mL) was added to the sample of vegetal extract (1 mL) and the mixture was incubated for 15 min at room temperature. After that, the absorbance values were read at 510 nm using a Shimadzu UV-1280 UV-VIS Spectrophotometer (Shimadzu Corporation, Kyoto, Japan). The reference standards used for flavonoid quantification were quercetin (QE) and the results are given in μg QE/mL extract.

### 4.5. Determination of Antioxidant Activity Using DPPH and ABTS Methods

#### 4.5.1. 2,2-diphenyl-1-picrylhydrazyl Radical Scavenging Assay (DPPH)

The determination was performed on the bases of the method described by Grochowski [20], on which we made small changes. Therefore, 50 µL of sample was added to 150 µL of 2,2-diphenyl-1-picrylhydrazyl (DPPH) 0.004% methanol solution. After 30 min incubation at room temperature in the dark, the absorbance was read at 517 nm. DPPH radical scavenging activity was expressed as milligrams of Trolox equivalents (mg TE/mL extract).

#### 4.5.2. 2,2′-azino-bis(3-ethylbenzothiazoline) 6-sulfonic acid Radical Scavenging Assay (ABTS)

The determinations where performed based on Grochowski’s method [20] with minor changes. ABTS•^+^ was generated by mixing 7 mM 2,2′-azino-bis(3-ethylbenzothiazoline) 6-sulfonic acid (ABTS) solution with 2.45 mM potassium persulfate (1:1, *v*/*v*). The mixture was allowed to stand for 12–16 min in the dark at room temperature. In the beginning of the assay, ABTS solution was diluted with methanol to reach an absorbance of 0.700 ± 0.02 at 734 nm. Then, 30 µL sample was added to 200 µL ABTS solution and vigorously mixed. After 30 min incubation at room temperature, the absorbance was read at 734 nm. The ABTS radical scavenging activity was expressed as milligrams of Trolox equivalents (mg TE/mL extract).

### 4.6. Obtaining Emulsions with Dermatocosmetic Applications

Three types of commercial W/O emulsions (A, B and C) were prepared based on 1 mL of *Galium verum* extract (HTE: S/L = 1:20/solvent concentration = 50%). These emulsions were differentiated by the chemical composition of the bases used (emulsifier, co-emulsifiers, and stabilizer), the general components of which are characterized in Table 6. The choice of the three main ingredients in the composition of the bases A, B and C was made in order to increase the performance of the product through a high permeability and to ensure a pleasant texture and optimal sensory characteristics for the patient/customer.

To obtain the final emulsion a protocol described by us previously was used [17] according to which the oily (A) and aqueous (B) phases were heated to 75 °C, after which the oily phase (A) was added over the aqueous phase (B). The mixing of the phases was done using a rotor-stator homogenizer (ESGE Zauberstab M 160 G Gourmet) which operates at 15,000 rpm. The obtained mixture was cooled to 40 °C (on an ice bath) after which the active ingredients and preservative (phase C) were added. HA oligo and HMW were finally added as a gel obtained by dissolving Hamamelis flowers in water. For further studies on emulsions, 15 g samples were weighed and packed in stained glass containers of adequate capacity. The samples thus prepared were kept in cool rooms until the analysis, for a maximum period of 90 days.

### 4.7. Preliminary Emulsions Characterization

The emulsions were preliminarily characterized by a series of analyzes to highlight the organoleptic characteristics and stability under certain conditions [13,14,15]. Thus, the organoleptic assessments, the determination of the pH, the separation of the phases under the action of centrifugal or vibrational force, the determination of the conductivity were followed. Prior to any analysis, the samples were allowed to return to room temperature.

#### 4.7.1. pH Determination

For determination of the pH values of dermato-cosmetic emulsions a digital pH Meter (Hanna Instrument) was used, and the protocol was: 0.5 g of emulsion was dissolved in 50 mL distilled water and was left to rest for two hours. The pH value was measured by inserting the electrode directly into the sample solution at 24.0 ± 2.0 °C. Following dilution with distilled water for pH test, emulsions presented a milky aspect and remained homogenous.

#### 4.7.2. Phases Separation

For centrifugation the test was performed using 5 g of sample which were introduced into a model XC-Spinplus centrifuge for 30 min at 25 °C and 3000 rpm. In the case of vortex test, it was used a Multi Speed Vortex MSV-3500 (Grant Instruments Ltd. Cambs, England), 5 g sample of emulsion and the experimental conditions were: 30 min at 25 °C and 3000 rpm.

#### 4.7.3. Microscopic Images

Microscopic images were taken from samples of the emulsions prepared after 7 days of storage under normal temperature conditions, using a binocular microscope Optika B-159 (OPTIKA S.r.l., Ponteranica (BG)—Italy), magnification—1000×.

#### 4.7.4. Conductivity Measurements

The conductivity measurements were performed with a portable Hanna Instruments type conductometer, on emulsion samples stored at an ambient temperature of 25 °C for 20 days.

## 5. Conclusions

In terms of antioxidant content (determined as flavonoids and total polyphenols), *Galium verum* extract proves to be satisfactory and corresponds to the idea of replacing chemical assets with natural ones in the formulation of certain dermato-cosmetic preparations.

The introduction of *Galium verum* extract in the preparation of a dermato-cosmetic emulsion allows us to obtain a preparation with a fine texture that is easily absorbed by the skin and the components of which are stable from a mechanical, physical, and biocompatibility point of view. These results encourage the development of further tests on rheology, microbiological stability, in vivo tests, and preliminary dermatological tests for these preparations.

## Figures and Tables

**Figure 1 plants-11-02454-f001:**
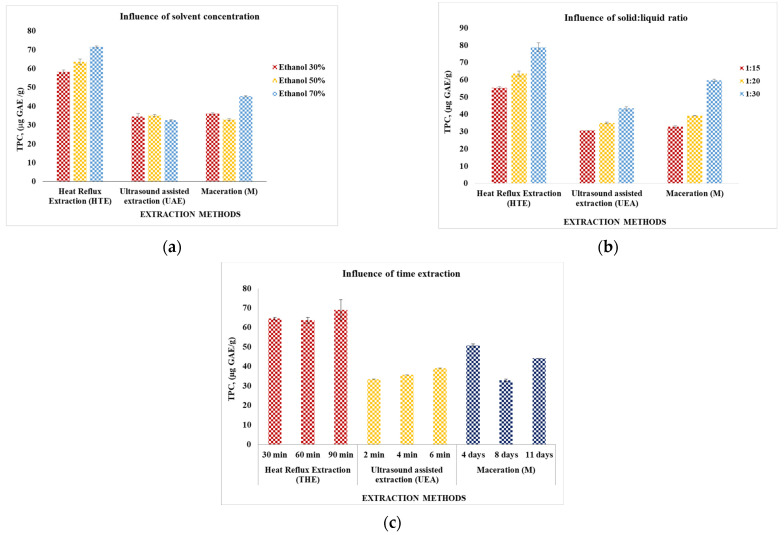
The total polyphenols (TPC) compounds content (μg GAE/g) of vegetal extracts obtained by *Galium verum* dry plant depending on the extraction method used and the physical parameters considered. Conditions: (**a**) HTE—solid/liquid ratio = 1:20; 60 min; M— solid/liquid ratio = 1:15; 8 days and UAE— solid/liquid ratio = 1:20; 4 min; (**b**) solvent concentration = 50%; HTE—60 min, M—8 days and UAE —4 min; (**c**) solvent concentration = 50%; HTE— solid/liquid ratio = 1:20; M— solid/liquid ratio = 1:15; and UAE—solid/liquid ratio = 1:20.

**Figure 2 plants-11-02454-f002:**
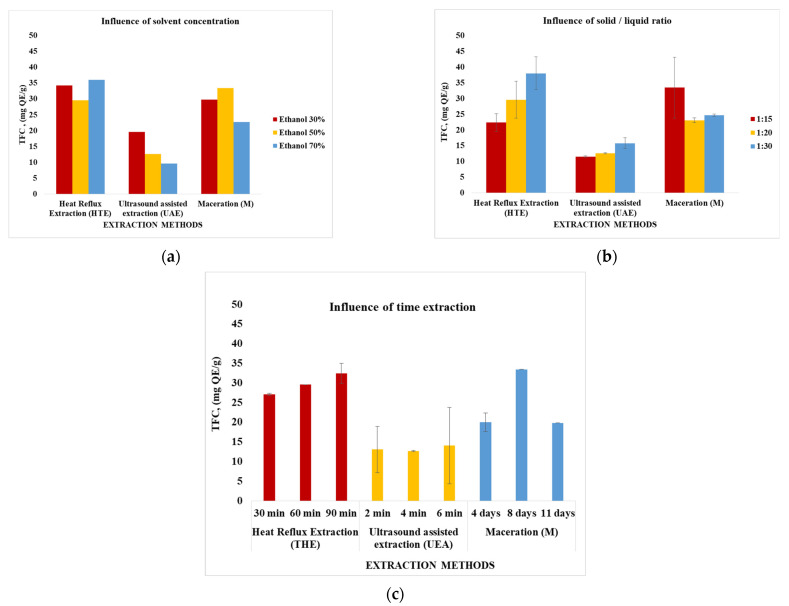
The total flavonoids compounds (TFC) content (mg QE/g) for vegetal extracts obtained by *Galium verum* dry plant depending on the extraction method used and the physical parameters considered. Conditions: (**a**) HTE—solid/liquid ratio = 1:20 and extraction time-60 min; M—solid/liquid ratio = 1:15 and extraction time -8 days; UEA- solid/liquid ratio = 1:20 and extraction time-4 min; (**b**) THE—solvent concentration = 50% and extraction time—8 days; UAE—solvent concentration = 50% and extraction time—4 min; (**c**) HTE- solvent concentration = 50% and solid/liquid ratio-1:20; M- solvent concentration = 50% and solid/liquid ratio-1:15; UEA—solvent concentration = 50% and solid/liquid ratio-1:20.

**Table 1 plants-11-02454-t001:** Biological activity of *Galium verum* extract.

Indications	Pathology	Biological Activity	Ref.
Internal use	Urinary stone complaints, Scurvy, Dropsy, Hysterics, Epilepsy, Gout, Nose bleeding, Stomach problems, Diarrhea, Scorbutic, scrofulous and dropsically complaints, Antistress—immunomodulatory, Pyelitis, Cystitis, Liver disorders, Cardiovascular diseases, Thyroidian, ovarian, adrenal and glucocorticoid hormones synthesis stimulation (in vitro).	Antioxidant (investigated and proved by DPPH, ABTS in vitro assays), Cytotoxic (investigated and proven in high doses in neck cancer cell lines HLaC78 and FADU: cell motility and invasion strong inhibition, DNA protection against benzo[a]pyrene’s toxicity in cigarettes), Protective: hepatic-protector, thymus protector, Antimicrobial (antibacterial and antifungal activity) (water, alcohol, chloroform extract): *Staphylococcus aureus, Escherichia coli, Pseudomonas aeruginosa, Bacillus subtilis, Proteus vulgaris*, *Candida albicans*, Endocrine system: morphological changes of hypothalamus-hypophysis-adrenal axis resulting in enhanced of neurosecretory activity, Antihaemolytic activity, Cholinesterase activation, Non-specific defense mechanism Detoxicant, Antibacterial, antifungal against Gram-positive microorganisms (*S. aureus*, *L. monocytogenes*) Anticandidiasis effect (on *Candida albicans*, *C. tropicalis*, *C. glabrata*)	[1,9,11]
External use	Indolent tumors, Strumous swelling and tumors of breast, Psoriasis, Delayed wound healing, Cancerous ulcer, Breast cancer, Bacterial and fungal infections, Parasitoses, Gingival inflammation, Cosmetic purposes		[1,2,7,12]

**Table 2 plants-11-02454-t002:** Degree of extraction obtained in liquid-solid extraction methods used, in the case of the fresh *Galium verum* plant.

Extraction Method	Sample	Characteristics			Extraction Degree, %
		Extraction Time	Ratio S:L	Solvent Concentration (%)	
Maceration (M)	M1 cr	8 days	1:15	30%	9.57
	M2 cr			50%	10.46
	M3 cr			70%	10.70
	M4 cr	8 days	1:20	30%	11.55
	M5 cr			50%	10.31
	M6 cr			70%	11.95
	M7 cr	8 days	1:30	30%	12.00
	M8 cr			50%	12.57
	M9 cr			70%	14.28
	M10 cr	4 days	1:15	50%	9.81
	M11 cr	11 days	1:15	50%	11.31
Heat Reflux Extraction (HTE)	R1 cr	60 min	1:20	30%	13.19
	R2 cr		1:20	50%	10.79
	R3 cr		1:20	70%	14.38
	R4 cr		1:15	50%	14.40
	R5 cr		1:30	50%	10.79
	R6 cr	30 min	1:20	50%	15.59
	R7 cr	90 min	1:20	50%	13.58
Ultrasound assisted extraction (UAE)	US 1 cr	4 min	1:20	30%	3.19
	US 2 cr			50%	2.80
	US 3 cr			70%	2.39
	US 4 cr	4 min	1:15	50%	1.49
	US 5 cr	4 min	1:30	50%	6.02
	US 6 cr	2 min	1:20	50%	3.19
	US 7 cr	6 min	1:20	50%	2.80

**Table 3 plants-11-02454-t003:** Degree of extraction obtained in liquid-solid extraction methods used, in the case of the dry *Galium verum* plant.

Extraction Method	Sample	Characteristics			Extraction Degree, %
		Time	Ratio S:L	Solvent Concentration (%)	
Maceration (M)	M1 sz	8 days	1:15	30%	26.59
	M2 sz			50%	26.59
	M3 sz			70%	22.11
	M4 sz	8 days	1:20	30%	28.34
	M5 sz			50%	28.29
	M6 sz			70%	27.54
	M7 sz	8 days	1:30	30%	29.31
	M8 sz			50%	28.06
	M9 sz			70%	27.43
	M10 sz	4 days	1:15	50%	34.40
	M11 sz	11 days	1:15	50%	33.75
Heat Reflux Extraction (HTE)	R1 sz	60 min	1:20	30%	33.59
	R2 sz	60 min	1:20	50%	33.19
	R3 sz	60 min	1:20	70%	34.38
	R4 sz	60 min	1:20	50%	31.82
	R5 sz	60 min	1:30	50%	31.82
	R6 sz	30 min	1:20	50%	33.60
	R7 sz	90 min	1:20	50%	32.77
Ultrasound Assisted Extraction (UAE)	US 1 sz	4 min	1:20	30%	15.18
	US 2 sz			50%	14.77
	US 3 sz			70%	12.30
	US 4 sz	4 min	1:15	50%	13.80
	US 5 sz	4 min	1:30	50%	16.19
	US 6 sz	2 min	1:20	50%	15.2
	US 7 sz	6 min	1:20	50%	17.6

**Table 4 plants-11-02454-t004:** TPC, TFC and antioxidant activity of investigated extracts.

Sample	DPPH (mg TE/mL)	ABTS (mg TE/mL)
R3 cr	3.68 ± 0.11	7.62 ± 0.39
R3 sz	12.19 ± 0.28	28.25 ± 0.88

The determinations were made in triplicate and the data are presented as mean ± standard deviation (SD). Abbreviations: ABTS—2,2′-azino-bis(3-ethylbenzothiazoline) 6-sulfonic acid; DPPH—1,1-diphenyl-2-picrylhydrazyl.

**Table 5 plants-11-02454-t005:** Preliminary Characterization of Emulsions with *Galium verum* Extract.

Parameters	Emulsions
A4	B4	C4
The appearance of the emulsion after centrifugation	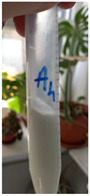	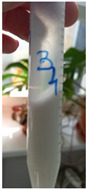	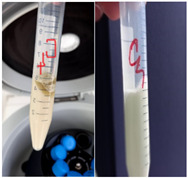
the appearance of the emulsion after vortex test	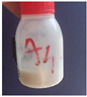	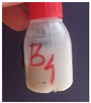	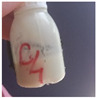
Stability after centrifugation and vortex	Intact texture and appearance	Thin upper layer of light foam	After first centrifugation—thin upper layer of green liquid due to phase separation, which led to conclusion that the emulsion could be unstable. After centrifugation in day 7 from preparation, the emulsion is intact, with stable texture.
pH (initial/after 7 days/after 14 days)	5.002/5.1/5.1	5.001/5.4/5.2	5.003/5.9
Organoleptic analyze	Color: bright white, slightly shiny.Texture: homogenous, firm, light/non-greasy, with no lumps detected after more than 72 h.Weak odor specific to emulsifier and floral water.	Color: bright white, slightly shiny.Texture: homogenous, firm, light/non-greasy, higher viscosity than A2 and C2, with no lumps detected after more than 72 h.Weak odor specific to emulsifier and floral water	Color: bright white, slightly shiny.Texture: initial -homogenous, firm, light/non-greasy, with no lumps detected after more than 72 h.Weak odor specific to emulsifier and floral water.
Microscopic image after 7 days of storage at 25 °C	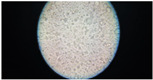	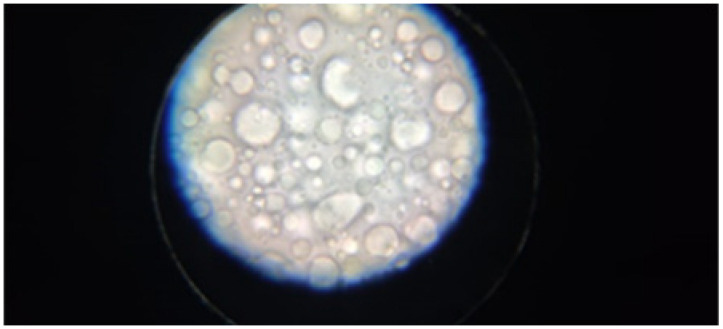	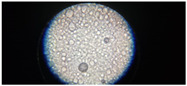
Conductivity measurement, mS after 7/21 days of storage at 25 °C	0.27/0.31	0.13/0.15	0.36/0.37

**Table 6 plants-11-02454-t006:** The main ingredients for the formulation of emulsions.

Phase	Role
Emulsifier	- 3 different compounds, for 3 different emulsions - versatile and flexible O/W, - superior sensory profiles, - pleasant appearance and lightweight texture
Co-emulsifiers	- supports stability, - increases their viscosity and gives an unctuous but also sliding texture, - helps other ingredients penetrate the skin better
Aqueous phase	floral waters
Stabilizer	- gelling agent - helps to stabilize emulsified creams and various cosmetic preparations - increases the viscosity of cosmetic compositions that also contain water
Preservatives	ensures the efficient preservation of a wide spectrum of cosmetic preparations: creams, lotions, toners, gels, hygiene products. It is an economical preservative, easy to use and approved for use in organic cosmetics
Active ingredients	vegetal extracts

## Data Availability

Not applicable.

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
