# Peer review of "Antioxidants from Galium verum as Ingredients for the Design of New Dermatocosmetic Products"

_plants, 2022, doi:10.3390/plants11192454_

Round 1
Reviewer 1 Report
The paper is interesting. However, the lack of statistical analysis of data disqualified the publication of manuscript in this form. Authors only included the standard deviations. The analysis of variance should be performed of other tests to show the statistically significant differences between means. Moreover, the number of repetition of individual test should be included.
In addition, the discussion of the results is very poor and almost not exists.
Lines 46-48: Please insert the citation.
Lines 164-182 The conclusion should be included in the end of the manuscript. In this chapter the results should be described and discussed.
Lines 400-401: Authors did not study how the skin absorb the emulsion. Moreover, this section is to overall and should include more conclusion form the performed study.
Reviewer 2 Report
The manuscript by Suteu and coworkers describes the extraction of antioxidants from Galium verum as well as the preliminary characterization for the emulsions, which contain one of the extracts. The experimental procedures have been carried out in a reliable way and the comparison of the extraction methos reported in this study would provide valuable information to the readers of Plants. Several points as indicated below need to be addressed by the authors before publication in the journal.
1. The title (line 2) is grammatically incorrect. “ingredients for design a new…” should be “ingredients for design of a new…”
2. What is the main component (polyphenol compound) of the extracts? Could the authors identify it?
3. The authors claim that the emulsions are stable. In that case, are they effective for the prevention of oxidative stress? The effective antioxidants are relatively unstable due to the autoxidation.
4. Line 36: “Gallium” should be “Galium.”
5. Line 53: “mong” should be “Among.”
Reviewer 3 Report
The manuscript by Turcov D. et al. is devoted to the comparison of different methods of extraction of Galium verum to obtain an extract suitable for production new dermatocosmetic products. The planning and execution of experimental work looks extremely strange. What was the basis for choosing the duration of the extraction? Why was 11 days chosen for miceration and 6 minutes for ultrasonic treatment? It is necessary to add information to the manuscript justifying such a choice. Many of the discussions in the manuscript are based specifically on comparisons of extraction efficiency, and the results of the efficiency evaluation do not appear to be correct.
It looks very strange to compare the extracts of the original and dried plants. If you want to compare the content of polyphenols, flavonoids, extraction efficiency, etc., then give the calculation for a completely dry sample of the plant.
The article does not correspond to the level of the journal "Plants" and cannot be accepted for publication in its current form.
Round 2
Reviewer 1 Report
The authors corrected the manuscript. However, the lack of statistical evaluation of data disqualified this manuscript. The explication of the authors is not satisfactory.
Reviewer 3 Report
I would like to thank the authors for their answers and comments. In the first version of the manuscript, I realized that the purpose of the work was to obtain decmatocosmetic emulsions. However, if the results of comparison of different extraction methods are presented, then it is necessary to clarify those aspects that raise questions. The authors referred to works 16 and 17, but these works also do not explain why such an extraction duration was chosen. For example, Figure 1c demonstrates that 6 min of ultrasonic extraction is not optimal, and the recovery rate continues to increase. I would still like to get an explanation for this protocol of the experiment, what was the purpose of the extractions pursued by the authors? Get extracts with high antioxidant activity?
This part of the experiment raises the main questions. I hope the authors can clarify this point and revise the manuscript.
Round 3
Reviewer 1 Report
I hope that in the next papers the authors will have a better approach to statistical evaluation of data.
Reviewer 3 Report
The authors answered my questions and substantiated the choice of extraction conditions. Probably, the scientific team has reason to use just such protocols. I hope that in further studies by the authors, these comments will be taken into account and there will be no questions when reviewing their manuscripts.